# Continuous generation of versioned collections' members with RML and LDES

Dylan Van Assche ⓘ, Sitt Min Oo ⓘ,
Julián Andrés Rojas ⓘ, and Pieter Colpaert ⓘ

IDLab, Department of Electronics and Information Systems,
Ghent University – imec, Technologiepark-Zwijnaarde 122, 9052 Ghent, Belgium
{dylan.vanassche, x.sittminoo,
julianandres.rojasmelendez, pieter.colpaert}@ugent.be

**Abstract.** When evolving datasets are used to generate a knowledge graph, it is usually challenging to keep the graph synchronized in a timely manner when changes occur in the source data. Current approaches fully regenerate a knowledge graph in such cases, which may be time consuming depending on the data type, size, and update frequency. We propose a continuous knowledge graph generation approach that can be applied on different types of data sources. We describe continuously updating knowledge graph versions represented as a Linked Data Events Stream, and use an RML processor for RDF generation. In this paper, we present our approach and demonstrate it on different types of data such as bike-sharing, public transport timetables, and weather data. By describing entities with unique, immutable, and reproducible IRIs, we were able to identify changes in the original data collection, reducing the number of materialized triples and generation time. Our use-cases show the importance of mechanisms to derive unique and stable IRI strategies of data source updates, to enable efficient knowledge graph generation pipelines. In the future, we will extend our approach to handle deletions in data collections, and conduct an extensive performance evaluation.

## 1 Introduction

Knowledge graphs (KG) usually coexist and integrate with heterogeneous and non-RDF data sources via mapping rules. The result of executing such mapping rules is commonly a data dump that represents a snapshot of the KG at a certain time. However, in practice, KGs and their sources may evolve making it difficult for data publishers to keep the KG up to date and for consumers to stay synchronized with their latest version. Recently, Linked Data Event Stream (LDES)[1] was introduced as a mechanism for publishers to expose a stream of changes (additions, updates and deletions) of topic-based data sources composed by data objects (a.k.a. members), as an immutable collection [1]. Thanks to LDES, data consumers are able to traverse and materialize historical versions, or simply synchronize with the latest member changes of a data source without having to

---

[1] https://w3id.org/ldes/specification

re-download it in its full extent, saving time, bandwidth, and computing resources.

Efficient generation of an LDES from non-RDF sources requires not only the capacity to define unique and immutable IRIs identifying every new version of a member, but also a mechanism that enables materializing only the members that were actually updated. Data sources may publish updates in various ways e.g. latest version of each member, only the updated members, or all members and their history. Yet, they not always provide the means to uniquely identify every update and leave it up to the consumer to handle integration.

Current KG generation approaches cannot leverage data updates. Instead, they regenerate the complete KG upon updates, even when only a subset of the data changes. This approach becomes impractical for large data sources where the cost of processing new updates grows proportionally to their update frequency and may impact services relying on the resulting KG. For example, services operating on outdated data or being unavailable during the regeneration.

We address this problem by investigating how we can uniquely and consistently (across multiple versions) identify and materialize only member updates, to avoid the costs of fully regenerating the KG. We extend an RML processor implementation to produce LDES-based KGs and demonstrate our approach through 3 use cases with varying data characteristics, e.g. size and update strategy. Preliminary tests show that our approach reduces the processing time and number of triples up to 20 times when a data source is updated compared to completely regenerating all the triples.

Through this work, we aim on reducing the generation time and size of the KG generated from dynamic and evolving data sources, while improving the overall operational behavior of dependent services and applications.

The next section discusses related work. Section 3 describes our approach. Section 4 applies our approach on 3 use cases: bike-sharing, public transport timetables, and weather data. Section 5 presents our conclusions and future work.

## 2   Related Work

*Mapping rules execution optimizations* Optimizing mapping rules execution for RDF KG generation is an active research domain; several approaches e.g. SDM-RDFizer [2], or FunMap [3] emerged to optimize the mapping rules execution for generating RDF. SDM-RDFizer avoids generating RDF from duplicates in a data collection. This way, all RDF triples are generated only once every time the SDM-RDFizer is executed. FunMap applies the same methodology to data transformations. It executes every function on each collection member only once and leverages an existing RML processor for the actual RDF generation. However, both approaches cannot apply this methodology across multiple versions of a data source. If a subset of a KG must be updated, it is fully regenerated.

*Linked Data Event Stream (LDES)* LDES[2] is a collection of immutable RDF data members such as versioned entities, observations, etc. LDES uses immutable IRIs [4] for each member. Consequently, every version of a member will have a new immutable IRI. LDES relies on the TREE specification [3] for describing collection and pagination relations. Each LDES has a `tree:shape` predicate which associates a SHACL [4] shape describing its members. It also includes `tree:member`s indicating the members of the collection. Optionally, one or multiple path descriptions such as `ldes:timestampPath` or `ldes:versionOfPath` may be provided to indicate the relation with previous versions of a member. Additionally, an LDES may specify a retention policy such as `ldes:LatestVersionSubset` to inform consumers for how long is the data kept.

*Versioning* Mechanisms to store and access versioned RDF data include the approaches proposed by Ostrich [5], TailR [6], R&Wbase [7], Memento[5], or LDES [8]. Three main RDF archive storage strategies can be identified: (i) Independent Copies (IC), (ii) Change-Based (CB), and (iii) Timestamp-Based (TB). Ostrich and TailR are both hybrid approaches, combining all three strategies for efficient query operations. R&Wbase is based on Git[6] and applies a CB approach, while Memento is a TB approach for accessing different versions over HTTP. All these approaches put the burden of resolving versions on the producer. LDES uses an IC approach by storing complete and incrementally versioned objects for each change in an append-only log. This way, consumers can synchronize their collections with the producer, and resolve the versioning on the consumer-side. LDES is similar to the Copy and Log approach [9], but can be applied using any versioning strategy, not only limited to a timestamp-based approach. However, efficient generation and storage of LDES-based KGs from dynamic non-RDF data sources remains an open challenge.

## 3 Approach

In this section, we describe our approach to generate unique and reproducible IRIs for collection's member updates, to enable continuous LDES-based KG generation. We discuss data collections types, how we generate IRIs, and expose a LDES.

### 3.1 Data collection types

Different data collection types exist regarding immutability and history. We identify 5 types given history availability and data immutability (Table 1).

Based on their update strategy, we consider that data collections may be: (i) *immutable* or (ii) *mutable*. A collection is *immutable* if all its members and their

---

[2] https://w3id.org/ldes/specification

[3] https://w3id.org/tree/specification

[4] https://www.w3.org/TR/shacl/

[5] http://mementoweb.org/guide/rfc/

[6] https://git-scm.com

updates can be uniquely identified through data member properties e.g. timestamps or hash codes. This way, consumers can identify each update individually without comparing against previous versions. In contrast, *mutable* collections require consumers to compare each member against previous versions to identify if the member was updated. In which case, consumers store the new version for handling later version reconciliation.

Regarding history availability, we identify 3 different types of data source: (i) *latest state*, (ii) *latest changes*, and (iii) *full history*. *Latest state* collections publish the latest version of all its members on every update. *Latest changes* collections publish only updated members (a.k.a. delta updates). In this case, consumers must retrieve the complete collection upfront and reconcile updates over it. In both previous cases, it is up to the consumers to record history. *Full history* collections provide the latest state of all members, including also (some) previous versions. In such case, member versions are normally identified by means of a version ID property, making them *immutable*.

| History | Immutable | Description | Use case |
|---|---|---|---|
| Latest state | Yes | Complete dump, unique IDs | Bike-sharing data |
| Latest state | No | Complete dump, non-unique IDs | GTFS timetables |
| Latest changes | Yes | Delta updates, unique IDs | OpenStreetMap diffs |
| Latest changes | No | Delta updates, non-unique IDs | GTFS-RealTime |
| Full history | Yes | Complete dump, versions history | Meteorological data |

**Table 1.** Data source types and example use cases according to their update strategy and characteristics.

### 3.2   Unique and reproducible IRIs

In this section, we introduce our approach to generate unique and reproducible IRIs by leveraging properties of *immutable* data collections or by observing properties of *mutable* ones.

*Immutable* data collections provide member properties which are renewed when a member is updated e.g. timestamps or hash codes. When generating KGs from this type of data collections, we use such properties to create unique and reproducible IRIs for named entities. We highlight that these properties must be unaffected by external factors. For example, timestamps may be affected by timezones, in such case, the timezone offset must also be present for the collection to remain *immutable*. *Mutable* data collections on the other hand, do not provide unique member properties across updates. Consumers must compare each data member against a previous version to identify updates. Once a member update is found, a unique property such as a timestamp or hash code needs to be externally generated to create unique and reproducible IRIs for named entities. Keeping track of the generated IRIs for each collection's member and updates, allows to identify updates and avoid generating duplicates.

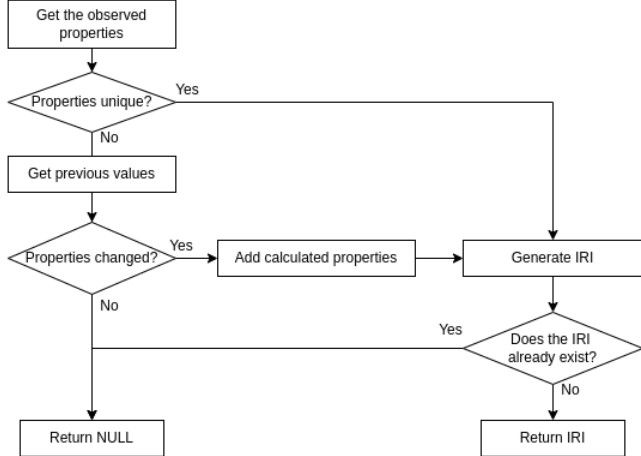

**Fig. 1.** Logical flow for generating unique and reproducible IRIs, when processing collection updates, and dealing with both *immutable* and *mutable* collections. We implement this logic as a FnO described function that is executed by the RML engine.

Our approach for generating unique and reproducible IRIs for mutable and immutable data collections is illustrated in Figure 1. We generate IRIs with an unique identifier for each data member update [10]. First, we check if the collection is immutable or not, which is described in the RML mapping rules. If immutable, we proceed to generate an IRI given that members already include at least one property to produce an IRI that is unique and reproducible (Section 3.1). If the collection is mutable, we compare the data member properties with a previous version (if any) to identify if a member was updated. To achieve this, we describe in the mapping rules which properties we want to observe of each data member. If a change is detected, we generate a unique value, e.g. timestamp or hash code based on the observed properties, to uniquely identify this member update and store this new version for comparison with further updates. Leveraging these unique and reproducible IRIs, we can avoid generating duplicates by checking if the IRI was already generated in the past or not. This way, only data members which are updated, trigger RDF materialization.

Listing 1.1 shows an extract of an RML mapping that produces unique IRIs for an *mutable* data collection. Subject IRIs are generated through a FnO described function (line 6). This function receives as input (i) the IRI template (line 9); (ii) a flag indicating if at least one property of the member is unique across updates (line 12) and; (iii) a set of properties to observe to check for updates (line 15). In this example, watched properties are input in a URL parameter-like form, but this can be adjusted according to the `idlab-fn:generateUniqueIRI` function implementation. Resulting subject IRIs in this example will conform to `http://ex.org/prop1/prop2#timestamp`. We use the generation time as times-

tamp in this example to make the IRI unique, but hash codes can be applied here as well.

**Listing 1.1.** RML Subject Map for generating a unique and immutable IRI from a *mutable* data collection.

```
1
2  rr:subjectMap [
3      fnml:functionValue [ # Unique IRI generation: $prop1/$prop2#timestamp
4          rr:predicateObjectMap [
5              rr:predicate fno:executes ;
6              rr:objectMap [ rr:constant idlab-fn:generateUniqueIRI ] ;
7          ], [ # IRI template
8              rr:predicate idlab-fn:iri ;
9              rr:objectMap [ rr:template "https://ex.org/{prop1}/{prop2}" ]
10         ], [ # Flag to indicate if the properties are unique on their own
11             rr:predicate idlab-fn:unique ;
12             rr:objectMap [ rr:constant "false"; rr:termType xsd:boolean; ]
13         ], [ # Set of properties to monitor for changes
14             rr:predicate idlab-fn:watchedProperty ;
15             rr:objectMap [ rr:constant "prop1={prop1}&prop2={prop2}"; ]
16         ];
17     ];
18 ]
```

### 3.3  LDES Logical Target

We also introduce LDES Logical Target which is used to export the generated LDES-based RDF into a file. Thanks to the modularity of RML's Logical Target [11], it is possible to add an LDES Logical Target to any RML processor.

An LDES Logical Target (Listing 1.2) is a regular RML Logical Target with an additional `ldes:EventStreamTarget` type. Each `ldes:EventStreamTarget` adds LDES specific properties to a regular Logical Target such as LDES paths (lines 6-7) or a SHACL shape (line 8). An LDES Logical Target only adds metadata to the serialized output. It also inherits all the characteristics of a regular RML Logical Target[7], allowing it to be written into a file, a triple store, etc.

**Listing 1.2.** LDES Logical Target definition in RML to export an LDES to a file.

```
1  <#LDESToFile> a rmlt:LogicalTarget;
2    rmlt:target [ a ldes:EventStreamTarget;
3      dcat:distribution [ a dcat:Distribution;
4        dcat:accessURL <file://data/dump.nq.zip>; ]; ];
5    rmlt:serialization formats:N-Quads; rmlt:compression comp:zip;
6    ldes:timestampPath sosa:resultTime;
7    ldes:versionOfPath dcterms:isVersionOf;
8    tree:shape <http://example.org/shape.shacl>; .
```

## 4   Use Cases and Preliminary Tests

In this section, we describe practical examples of our approach, applied on 3 use cases: (i) BlueBike[8] bike-sharing availability data, (ii) NMBS[9] public transport

---

[7] https://rml.io/specs/dataio/

[8] https://blue-bike.be/

[9] https://www.belgiantrain.be/

timetables, and (iii) KMI[10] meteorological data with the RMLMapper as RML processor. We implemented our approach in the RMLMapper, but it can be implemented in any other RML processor since we re-use existing features such as FnO functions to generate unique and reproducible IRIs; and RML's Logical Target for producing LDES-based KGs. Our mapping rules [11] and implementation [12] are available on GitHub.

*BlueBike* bike-sharing data provides a dump of all stations and currently available bikes. This data source is *immutable*. Its members include a data property that is unique across updates, and it follows a *latest state* update strategy. We mapped the data using a GBFS-based vocabulary [13] into RDF, and exported it as an LDES with our LDES Logical Target. We retrieved every minute the latest state and generated unique and reproducible IRIs with our approach (Section 3.2), by leveraging the immutable last_seen timestamp property, which is included on every data member of this collection.

*NMBS* is the Belgian railway agency which publishes its yearly timetable schedule as a GTFS [14] dump. This data source is *mutable* since there are not unique data properties across updates, and follows a *latest state* update strategy. Each day the schedule is republished without providing a list of updates. The GTFS dump is retrieved daily to check for updates. We define a set of observed properties that are used to asses member updates (e.g. parent_station and stop_id for GTFS stops). If an updated member is found, we generate a unique and reproducible IRI for the member by adding the current timestamp when the mapping rules were executed.

*KMI* is the Belgian meteorological institute, which provides measurements of their Automatic Weather Stations as a CSV dump. This data source is *immutable* since it contains historical data about the measurements with unique identifiers. We leveraged these identifiers, namely the code property, to generate unique and reproducible IRIs.

We performed preliminary tests by retrieving dumps for each of the use cases and replayed them in the same retrieval order to verify the impact of our approach on generation time and number of materialized triples. We observed reduced generation time that is proportional to the data source size. For the smaller data sources BlueBike and KMI, the generation time is 1.1x and 1.2x faster. For the bigger data source NMBS we register a generation time 24.7x faster. In terms of number of materialized triples, we see a reduction in all cases: BlueBike 4.6x less triples, KMI 17x less triples, and NMBS 33.7x less triples; compared to fully regenerating each KG. These results are available on Github[15].

---

[10] https://opendata.meteo.be/

[11] https://github.com/RMLio/RML-LDES-mapping-rules

[12] https://github.com/RMLio/rmlmapper-java/releases/tag/v5.0.0

[13] https://github.com/pietercolpaert/Blue-Bike-to-Linked-GBFS

[14] https://gtfs.org

[15] https://github.com/RMLio/RML-LDES-mapping-rules/

## 5   Conclusion

We see very promising results of our approach in terms of efficiency for continuous KG generation. We manage to speed up generation time up to 24x while reducing unnecessarily materialized triples up to 33.7x. Our approach relies on existing RML features such as FnO functions, which facilitates its implementation on existing engines. Furthermore, our implementation provides *out-of-the-box* publishing of KG updates in the form of an LDES, which facilitates data synchronization tasks for consumers. In the future, we will extend our approach to handle deletions and perform a more exhaustive performance evaluation.

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
