# OpenReview forum: "Continuous generation of versioned collection's members with RML and LDES"
_kg-construct.github.io/KGCW/2022/Workshop — KGCW 2022_

### Official Review · ~Oscar_Corcho1 · 2022-03-26
**Nice ideas for discussion on a workshop. More details on the approach would have been good**

**Rating:** 6
**Confidence:** 4

**Review:**

Work on a relevant topic that stems from the real application of KGC in real domains, addressing the need  to perform incremental transformations of data sources, considering the different types of data sources that we may need to interact with in the process in relation to how the identification of changes in the data can be done and how depending on this the incremental generation process has to be approached.

For a workshop paper, I consider this a relevant contribution that can actually contribute to the discussion in the workshop, and hence I think that it would be nice to get it accepted and discussed. I especially like the analysis of different types of sources wrt how they handle updates, since it is useful to provide characterisations of them and hence apply potential solutions.

My "complain" about the paper is that it uses too much space for the motivation (which is very easy to grasp for those in the KGC workshop) and for the background and related work, which is probably not so relevant for the audience of this workshop, especially because then there is no comparison with how this problem is actually addressed by some of the existing solutions. And as a result, the approach description is actually very limited.
- I would have been interested in a bit more detail and statistics on data sources of each of the data types / groups described in section 4.1.  So that it is easier to determine where most effort should be devoted in the state of the art.
- I would have also appreciated a bit more detail on section 4.2. Indeed, I am not sure why this is called algorithm, since I understand that you base your decision on the analysis of the dataset and its attributes, and this is a very simple decision process. I have applied in the past approaches like the one of generating timestamps, etc. I wonder why not using hash codes, for example. Indeed, this discussion reminded me of very early and non-evaluated work that we did in 2009 on identifiers for Linked Stream Data (https://oa.upm.es/5442/).
- After the discussion on identifiers, I cannot really understand well whether you have to evaluate in some cases the whole dataset to make the diff or not, so as to determine whhether transforming something or not.
- The LDES Target seems ok to me. A syntactic aspect, but relevant.

In terms of showing how the approach can be used in different use cases, I would not call that section Validation, but Use Cases or alike. I would also have liked to see the generated RDF in different timestamps.

Finally, the authors refer to a GitHub repository, but I have not seen the URL in the paper. Maybe I missed it.

---

### Official Review · ~Samaneh_Jozashoori1 · 2022-03-29
**An approach for a common and important issue in materialized knowledge graph creation. Inadequate explanation of evaluation!**

**Rating:** 5
**Confidence:** 4

**Review:**

The paper presents an incremental approach for materializing knowledge graphs. It is claimed that the proposed approach promises efficiency while preserving completeness.

The paper is well-written, however, there are quite some typos, missing/wrong prepositions, etc that need to be revised.

The first three sections are very clearly and extensively explained; the work is well motivated. Maybe only in the abstract, it would be better to clarify from the beginning that by "current approaches" the authors mean the "materialized" knowledge graph approaches.

Approach: contrary to the previous sections, unfortunately, this section is not valued enough. I would appreciate an extensive explanation of different considered categories to better and more clearly understand. I'm also concerned that by mostly describing the approach only according to specific use-cases or examples, one may doubt the comprehensiveness of the approach.

Evaluation: considering the abstract: "... we were able to easily identify changes in the original data collection, reducing up to 20 times the amount of materialized triples and their generation time..." I was expecting to read a complete explanation of empirical evaluation setups that illustrate the results claimed in the abstract as well as the "Preliminary results" in the "evaluation" section but surprisingly that is not the case. I consider the lack of an adequate evaluation explanation and reproducibility (it is mentioned but the GitHub link is not provided) a major issue that is required to be solved before the presentation of a paper.

---

### Official Review · ~Christophe_Debruyne1 · 2022-03-30
**Great and relevant idea, but details about the validation are missing.**

**Rating:** 5
**Confidence:** 4

**Review:**

The authors proposed an approach to generate IRIs to represent changes in a knowledge graph's underlying data source. Depending on the nature of the source data, the extended RML engine can immediately focus on the changes or needs to compare two versions. Their proposal focuses on additions and updates, with deletions mentioned as future work. Their contribution is already valuable for contexts in which data is not deleted but only added (e.g., sensor data streaming). Overall, the paper is well-written, and its topic fits the workshop's scope and is topical. The presentation of the use cases and the preliminary results were not compelling because the paper did not provide sufficient information. The use cases did cover different scenarios, and I would have appreciated including a concrete example (of a mapping). The paper can be improved by "completing" the results section or replacing it with a demonstrator. The latter would make the paper a good position paper.

The authors mentioned use cases and preliminary results in their validation section. The GitHub link was not provided, preventing me from looking at some details which left unanswered, such as:

- How does one declare which properties to observe?
- Were vocabularies extended or is this an application of existing vocabularies with a bespoke RML engine? From Section 1, it seems the latter. A concrete example would have made this clear.

Even though the results are preliminary, I would have appreciated it if the authors explained the figures in Section 5.2. The paragraph only contains two sentences, with the first being a summary. Are there only three factors (5, 10, and 20) for each data source? Have you run these mappings on several occasions, and are these factors averages?

Final thought
The abstract mentions a reduction in time, whereas the introduction expands on this by saying also computing resources. This work could thus contribute to reducing the carbon footprint of computing.

Clarifications and minor comments:
Define "differential" in differential knowledge generation.
What is meant by an "immutable IRI"? Do you have a definition? Do you refer to "immutable" as defined by [1]?
Can you provide a reference to the TREE specification?
Consider adding line numbers to listings.
To keep the paper as self-contained as possible, the authors could provide footnotes explaining GBFS and GTFS.
For some reason, all links in the document point to localhost:3000.

[1] Tobias Kuhn, Michel Dumontier: Trusty URIs: Verifiable, Immutable, and Permanent Digital Artifacts for Linked Data. ESWC 2014: 395-410

Text improvement:
Overall
Consistency: "Knowledge Graph" vs. "knowledge graph", "e.g.," vs. "e.g.", "bike sharing" vs "bikesharing", "public transit" vs. "public transport", "timetable" vs. "timetables"
Mixing of British English and American English: "modelled" is the former and "heterogenous" the latter.
If possible, it would be beneficial to have the listings and tables after their mention in the text (very minor).

Title
"... a versioned collection's ..." or "... versioned collections' ..."?

Abstract
"..., IT is usually challenging ..."?
"... applied on ..." instead of "... applied to ..."
"bike sharing" has no meaning. Do you mean bike-sharing data?
"20 times" instead of "20x" (same comment for Section 5)
"conduct" instead of "execute"

Section 1
"integrate with" -> KGs integrate with "what" ? Or do you mean "are integrated with" in this sentence?
A comma between "evolve" and "making" is missing.
"to its full extent" instead of "in its full extent"
Remove the comma between "... updates, to avoid ..."
"... and THE number of triples ... "
"... to reduce ..." instead of "... on reducing ..."
"timetables" instead of "timetable"
A comma is missing in the last sentence.

Section 2
"a KG" instead of "an KG"
"proposed" instead of "porposed"
"..., AND IS not only limited to..."?
"However" instead of "Yet"

Section 3
You can remove "to " in "... exported to" (multiple occurrences)
I do not see how the first line in Listing 2 relies on TREE.
A comma after "Additionally" is needed

Section 4
"a collection's member updates" or "collections' member updates"?
One usually puts table captions above the table.
"A unique" instead of "an unique" (multiple occurrences)
"... in any OF its properties..." ("of" is missing)
"choose not to publish" instead of "choose to not publish"
The last sentence of Section 4.1 misses a word. An "and thus" after the comma, perhaps?
"watching properties" vs. "observing properties" (Fig. 1 vs. the text)
"If one of these values differS"
In Section 4.2, you refer to line 9 in Listing 3, but there are only 7 lines. I don't see last_seen in the listing either.

Section 5
A comma after "In this section"
A link to the GitHub repository seems to be missing.
A comma between "our approach" and "given"

Section 6
"a performance evaluation" instead of "an performance evaluation"

References
Issues with last names in reference 10
Reference 1 is incomplete
Some references are incomplete, e.g., reference 7

---

### Official Review · ~Vladimir_Alexiev1 · 2022-04-01
**Good follow-on work, but not enough details on the contribution**

**Rating:** 5
**Confidence:** 4

**Review:**

This is a continuation of previous works on LDES and RML Targets, which is good since it develops these ideas further.

The paper proposes to compute RDF change streams from changing source data.
The change streams are limited to whole entities (changes are not at triple-level).
Entities are treated as immutable, and in a sense as "value objects". i.e. every change to important properties leads to a new subject IRI.

However, there are not enough details to evaluate the new contribution.
"Our mapping rules and implementation are available on GitHub" but no link is provided.

Questions that seem important to me but I couldn't answer from the paper:

* How do you define "the observed properties" to be compared to last known values? Is this limited to `tree:path` and `ldes:timestampPath`?
* How do you define the extent of a "change"? How do you "circumscribe" an entity, especially in cases where the entity shape is complex (includes triples not just 1 hop away from the subject). Is this somehow related to `tree:shape`?
* Or is change detection described completely in RML? Then how do you guarantee the RML is consistent with the LDES definition?
* How do you handle global change? Eg in the case of Railway schedules, what happens when the timezone changes (eg Daylight Savings Time hits)?
* Can ldes:EventStreamTarget write directly to a repository?
* Can ldes:EventStreamTarget write to Kafka, which is often used in enterprise scenarios
* Do you see any use for Named Graphs in these scenarios?
* When using a ldes:DurationAgoPolicy, which component takes care to purge older entities? Are they purged from a semantic repository, or a data dump?
* Reduction of 5-10-20x compared to what?

---

### Decision · Program_Chairs · 2022-04-11

**Decision:**

Accept

**Comment:**

Dear authors,

Considering the reviews received for the paper, we have decided to give a conditional acceptance for the paper, as the topic tackled is very relevant for the workshop but major changes are required. All reviews agreed that the contributions need to be more detailed, evaluation needs to be extended and better explained, Github repo must be provided to ensure reproducibility and text should also be improved (many typos and syntax errors).

We encourage the authors to take into consideration the reviews very carefully, solve the current issues, and upload a new version of the paper before the 25th of April. We will take the final decision based on that version that should be as close as possible to a potential camera-ready version. Optionally, the authors could provide (and post here as a comment) a document/letter explaining how they've resolved the required changes.

----------------------------------

The authors have updated the paper according to the comments from the reviewers, so now it is accepted for its presentation in the workshop.

---

> ### Author Response · Authors · 2022-04-25
> **Changes Part 1**
>
> # Changes to address comments of reviewers
> ​
> ## Reviewer 1: Oscar Corcho
> ​
> > My "complain" about the paper is that it uses too much space
> > for the motivation (which is very easy to grasp for those in the KGC workshop)
> > and for the background and related work, which is probably not so relevant for the audience of this workshop,
> > especially because then there is no comparison with how this problem is actually addressed by some of the existing solutions.
> > And as a result, the approach description is actually very limited.
> ​
>
> As highlighted by the reviewer, we can reduce the Background and Related Work sections because
> they are easy to grasp for the audience in the KGC workshop.
> We used this additional space to expand the Approach section.
> ​
>
> > I would have been interested in a bit more detail and statistics on data sources
> > of each of the data types / groups described in section 4.1.
> > So that it is easier to determine where most effort should be devoted in the state of the art.
> ​
>
> We expanded section 3.1 Data collection types further to better explain
> the different types of data collections we identified and how they differ from each other.
> ​
>
> > I would have also appreciated a bit more detail on section 4.2.
> > Indeed, I am not sure why this is called algorithm,
> > since I understand that you base your decision on the analysis of the dataset and its attributes,
> > and this is a very simple decision process.
> > I have applied in the past approaches like the one of generating timestamps, etc.
> > I wonder why not using hash codes, for example.
> > Indeed, this discussion reminded me of very early and non-evaluated work that
> > we did in 2009 on identifiers for Linked Stream Data (https://oa.upm.es/5442/).
> ​
>
> We acknowledge the reviewer's comment here, therefore we expanded Section 3.2 as well to clearify
> that any kind of unique property can be used.
> The approach is not limited to timestamps, so hash codes can be used as well.
> The reference mentioned by the reviewer is relevent and is added.
> ​
>
> > After the discussion on identifiers,
> > I cannot really understand well whether you have to evaluate in some cases the whole dataset
> > to make the diff or not, so as to determine whhether transforming something or not.
> ​
>
> We clarified further when we need to perform a comparision diff or not for each type of data collection.
> For immutable data collections, we do not need to perform a comparision diff,
> only for mutable data collections because they do not include any properties uniquely identifying each data update.
> ​
>
> > In terms of showing how the approach can be used in different use cases,
> > I would not call that section Validation, but Use Cases or alike.
> > I would also have liked to see the generated RDF in different timestamps.
> ​
>
> We renamed the section to 'Use Cases and Preliminary Tests' to address this comment.
> ​
>
> ## Reviewer 2: Samaneh Jozashoori
> ​
> > Approach: contrary to the previous sections, unfortunately, this section is not valued enough.
> > I would appreciate an extensive explanation of different considered categories to better and more clearly understand.
> > I'm also concerned that by mostly describing the approach only according to specific use-cases or examples,
> > one may doubt the comprehensiveness of the approach.
> ​
>
> We expanded the Approach section as mentioned above.
> Moreover, we moved the use-cases to the 'Use Cases and Preliminary Tests' section to discuss our approach better.
> ​
>
> > Evaluation: considering the abstract: "... we were able to easily identify changes in the original data collection,
> > reducing up to 20 times the amount of materialized triples and their generation time..."
> > I was expecting to read a complete explanation of empirical evaluation setups
> > that illustrate the results claimed in the abstract as well as the "Preliminary results"
> > in the "evaluation" section but surprisingly that is not the case.
> > I consider the lack of an adequate evaluation explanation and reproducibility
> > (it is mentioned but the GitHub link is not provided)
> > a major issue that is required to be solved before the presentation of a paper.
> ​
>
> We changed the abstract to better highlight that we mostly focus on verifying the comprehensiveness of our approach.
> Furthermore, we uploaded these preliminary results to GitHub (https://github.com/RMLio/RML-LDES-mapping-rules/)
> and made the link better visible
> by rewriting the paper in LaTeX instead of HTML which lacks proper highlighting for links in LNCS mode.

---

> ### Author Response · Authors · 2022-04-25
> **Changes Part 2**
>
> ## Reviewer 3: Christophe Debruyne
> ​
> > The presentation of the use cases and the preliminary results were not compelling because the paper
> > did not provide sufficient information. The use cases did cover different scenarios,
> > and I would have appreciated including a concrete example (of a mapping).
> > The paper can be improved by "completing" the results section or replacing it with a demonstrator.
> ​
>
> We acknowledge the reviewer's comment and addressed it in the Approach section by expanding it with a concrete example.
> ​
>
> > The authors mentioned use cases and preliminary results in their validation section.
> > The GitHub link was not provided, preventing me from looking at some details which left unanswered
> ​
>
> We rewrote the paper in LaTeX instead of HTML to better highlight links.
> All links to the mapping rules are now properly highlighted such as https://github.com/RMLio/RML-LDES-mapping-rules/.
> ​
>
> > Even though the results are preliminary, I would have appreciated it if the authors explained the figures in Section 5.2.
> > The paragraph only contains two sentences, with the first being a summary.
> > Are there only three factors (5, 10, and 20) for each data source? Have you run these mappings on several occasions,
> > and are these factors averages?
> ​
>
> We clarified the results in the Section 'Use Cases and Preliminary Tests'
> and provided a more detailed explanation and the measurements on Github: https://github.com/RMLio/RML-LDES-mapping-rules/.
> ​
>
> ## Reviewer 4: Vladimir Alexiev
> ​
> > How do you define "the observed properties" to be compared to last known values?
> > Is this limited to tree:path and ldes:timestampPath?
> > How do you define the extent of a "change"? How do you "circumscribe" an entity,
> > especially in cases where the entity shape is complex (includes triples not just 1 hop away from the subject).
> > Is this somehow related to tree:shape?
> > Or is change detection described completely in RML? Then how do you guarantee the RML is consistent with the LDES definition?
> ​
>
> We clarified this by adding an example to the paper.
> The observed properties are described in the FnO description in the RML mapping rules.
> The observed properties are compared to the latest value stored by the RML processor.
> ​
>
> > How do you handle global change? Eg in the case of Railway schedules,
> > what happens when the timezone changes (eg Daylight Savings Time hits)?
> ​
>
> We added an explanation in the Approach section to highlight that immutable data collections are only immutable
> if their unique properties are unaffected by such influences. If not, they are not considered immutable.
> ​
>
> > Can ldes:EventStreamTarget write directly to a repository?
> > Can ldes:EventStreamTarget write to Kafka, which is often used in enterprise scenarios
> ​
>
> Yes, LDES Logical Target inherits all characteristics of a regular RML Logical Target.
> RML Logical Target could write to Kafka, but this has not been implementing yet.
> However, this is being worked on in the RML Source Target specification: https://github.com/kg-construct/rml-target-source-spec
> ​
>
> > Do you see any use for Named Graphs in these scenarios?
> ​
>
> Named Graphs can be used with this approach, but they are not required by this approach for it to work.
> ​
>
> > When using a ldes:DurationAgoPolicy, which component takes care to purge older entities?
> > Are they purged from a semantic repository, or a data dump?
> ​
>
> This policy advertises to LDES clients how long the data is kept.
> If a client needs history but the LDES producer does not keep any,
> the client is informed about this and can keep the history locally.
> We provide this information to comply with the LDES specification.
> ​
>
> > Reduction of 5-10-20x compared to what?
> ​
>
> We clarified in 'Use Cases and Preliminary Tests' section that the reduction is compared against fully regeneration the KG.